# Ibrutinib Resistance Mechanisms and Treatment Strategies for B-Cell Lymphomas

**DOI:** 10.3390/cancers12051328

**Published:** 2020-05-22

**Authors:** Bhawana George, Sayan Mullick Chowdhury, Amber Hart, Anuvrat Sircar, Satish Kumar Singh, Uttam Kumar Nath, Mukesh Mamgain, Naveen Kumar Singhal, Lalit Sehgal, Neeraj Jain

**Affiliations:** 1Department of Hematopathology, MD Anderson Cancer Center, Houston, TX 77030, USA; bhawanantu@gmail.com; 2Department of Internal Medicine, the Ohio State University, Columbus, OH 43210, USA; Sayan.MullickChowdhury@osumc.edu (S.M.C.); Amber.Hart@osumc.edu (A.H.); Anuvrat.Sircar@osumc.edu (A.S.); Satishkumar.Singh@osumc.edu (S.K.S.); 3Department of Medical Oncology & Hematology, All India Institute of Medical Sciences, Rishikesh 249203, India; uttam.haemat@aiimsrishikesh.edu.in; 4Department of Biochemistry, All India Institute of Medical Sciences, Rishikesh 249203, India; Mukeshmamgain77@gmail.com (M.M.); naveen.bchem@aiimsrishikesh.edu.in (N.K.S.)

**Keywords:** ibrutinib, acquired resistance, genetic alterations, tumor microenvironment, BTK-PROTAC, CAR T-cells

## Abstract

Chronic activation of B-cell receptor (BCR) signaling via Bruton tyrosine kinase (BTK) is largely considered to be one of the primary mechanisms driving disease progression in B–Cell lymphomas. Although the BTK-targeting agent ibrutinib has shown promising clinical responses, the presence of primary or acquired resistance is common and often leads to dismal clinical outcomes. Resistance to ibrutinib therapy can be mediated through genetic mutations, up-regulation of alternative survival pathways, or other unknown factors that are not targeted by ibrutinib therapy. Understanding the key determinants, including tumor heterogeneity and rewiring of the molecular networks during disease progression and therapy, will assist exploration of alternative therapeutic strategies. Towards the goal of overcoming ibrutinib resistance, multiple alternative therapeutic agents, including second- and third-generation BTK inhibitors and immunomodulatory drugs, have been discovered and tested in both pre-clinical and clinical settings. Although these agents have shown high response rates alone or in combination with ibrutinib in ibrutinib-treated relapsed/refractory(R/R) lymphoma patients, overall clinical outcomes have not been satisfactory due to drug-associated toxicities and incomplete remission. In this review, we discuss the mechanisms of ibrutinib resistance development in B-cell lymphoma including complexities associated with genomic alterations, non-genetic acquired resistance, cancer stem cells, and the tumor microenvironment. Furthermore, we focus our discussion on more comprehensive views of recent developments in therapeutic strategies to overcome ibrutinib resistance, including novel BTK inhibitors, clinical therapeutic agents, proteolysis-targeting chimeras and immunotherapy regimens.

## 1. Introduction

BCR signaling and its component BTK have direct pathogenic roles in the development of multiple B-cell malignancies, including chronic lymphocytic leukemia (CLL), mantle cell lymphoma (MCL), diffuse large B cell lymphoma (DLBCL), follicular lymphoma (FL) and Waldenstrom’s macroglobulinemia (WM) [1]. BCR signaling is constitutively active in these B-cell malignancies through both ligand-dependent (triggered by antigens present in the lymphoma-associated tumor microenvironment; TME) and independent mechanisms (autonomous BCR stimulation by its epitope) leading to the activation of oncogenic NF-kB and other targeted survival signaling [2,3,4]. This chronically active BCR signaling promotes clonal proliferation and accumulation of malignant B-lymphocytes in the bone marrow, secondary lymphoid organs, and blood [5]. BTK is a central contributor to B-cell lymphoma pathogenesis, as it is significantly more active (increased phosphorylation) in B-cell lymphomas than in normal B-cells [6]. Additionally, BTK plays a fundamental role in modulating chemokine and integrin signaling (CXCR4), thereby regulating B-cell trafficking and tissue homing [7]. Therefore, therapies directed towards targeting BTK have proven their potential in both pre-clinical and clinical settings. The BTK inhibitor ibrutinib (Imbruvica, Pharmacyclics, LLC), a first-generation BTK inhibitor which showed evidence of high efficacy and acceptable toxicity profile in clinical trials, was the first drug in this category to receive FDA approval as a single-agent frontline therapy for the B cell malignancies MCL, WM and CLL (including patients carrying del(17p) or TP53 mutation) [8]. In addition to targeting BCR signaling, ibrutinib has appeared to control TME by regulating cytokine signaling, modulating activity of tumor-associated stromal cells and promoting redistribution of tumor cells in order to induce lymphocytosis and therapeutic-agent-dependent apoptosis [9,10]. Recent reports have investigated the effect of ibrutinib therapy on epigenomic changes and in upholding chimeric antigen receptor (CAR) T-cell therapy activity [11,12,13]. Despite encouraging preclinical results and initial clinical success, the response of lymphoma patients’ to ibrutinib is frequently found to be inadequate, with reports of drug-dependent toxicity and disease progression on discontinuation of drug therapy [14,15]. Additionally, activation of bypass mechanisms, clonal evolution, or acquired genetic alterations have led to the development of ibrutinib resistance in some patients [13,15]. Therefore, identification of novel targeting agents that function either alone or in combination with ibrutinib to provide better control of disease progression while reducing ibrutinib-associated toxicities is currently of high clinical significance. In this article, recent progress in genetic, non-genetic and other sources of ibrutinib resistance and advanced treatment combinations to overcome ibrutinib resistance have been reviewed.

## 2. Complexities of Ibrutinib Resistance Development

Despite the promising activity of ibrutinib across multiple B-cell lymphoma subtypes, almost one third of patients have primary intrinsic resistance, while many others appear to develop acquired resistance [1]. In primary drug resistance, patients do not respond to initial therapy, whereas in secondary drug resistance, patients have an initial response that is subsequently lost due to acquired resistance or clonal evolution. Understanding both primary and secondary resistance mechanisms is essential for the development of appropriate therapeutic strategies. Here, we have discussed the genetic and non-genetic foundations of primary and acquired ibrutinib resistance development in B-cell lymphoma.

### 2.1. Genetic Causes of Ibrutinib Resistance

Two studies have demonstrated the secondary cause of ibrutinib resistance development in CLL [16,17]. Furman et al. described a case report where the patient developed progressive disease following 21 months of ibrutinib treatment and initial positive response. RNA sequencing identified a mutation of *BTK* (C481S) which was not detected before ibrutinib therapy [16]. Woyach et al. performed exome sequencing at baseline (before start of ibrutinib treatment) and at the time of relapse on six CLL samples and identified BTK^C481S^ mutation in 83% (5/6) patients, and *PLCG2* mutation in 33% (2/6) patients, that were not in baseline samples [17]. Further studies demonstrated gain of function mutations in *PLCG2* (R665W, L845F, S707Y) that could be attributed to a secondary mechanism of ibrutinib resistance in CLL and WM [17,18,19,20]. These *BTK* and *PLCG2* mutations are rarely seen in MCL patients. Although acquired mutations in *BTK* or its downstream mediator *PLCG2* have been identified in the majority of ibrutinib-resistant cases (80%), not all patients progressing on ibrutinib harbor these alterations. Table 1 shows selected studies that define alternative gene mutations instead of common BTK or PLCG2 mutations associated with ibrutinib resistance development.

Next-generation sequencing has uncovered new genomic perturbations including sub-clonal heterogeneity, clonal shift, and patterns of clonal variegation during disease progression. Lymphoma cells under ibrutinib pressure are prone to clonal shift. Jimenez et al. recently identified homozygous loss of chr:6q or 8q in cases of MYD88^L265P^-mutant WM who progressed while on ibrutinib treatment [27]. It is important to note that critical negative regulators of BTK, MYD88/NF-κB and apoptotic signaling are located on chr:6q and 8q. Another clinical trial has identified an activating mutation of CXCR4 (S338X) which is commonly observed in nearly 30% of MYD88^L265P^ WM cases. Double mutant (MYD88^L265P^/CXCR4^S338X^) patients have been found to be associated with a lower response to ibrutinib [28].

Amin et al. performed whole-exome sequencing (WES) on paired 48 CLL samples and found that 5 out of the 48 samples were substantially more resistant to ibrutinib following relapse from chemotherapy. Three out of the five resistant samples had acquired a del17p/TP53 mutation [13]. Another group performed targeted sequencing for selected 29 genes on 11 paired CLL samples and further stratified acquired mutations in *TP53*, *SF3B1*, and *CARD11* after disease progression [29]. Other genetic mutations associated with ibrutinib resistance in CLL include *del(8p)*, which leads to TRAIL insufficiency in conjunction with driver mutations in *EP300*, *EIF2A*, and *MLL2* [33] and novel *BTK* mutation (BTK^T316A^) that induces ibrutinib resistance via activating PLCG2 in CLL [31,33].

Mutations in the *MYD88* gene (MYD88^L265P^) are among the most prevalent in B-cell lymphomas, including activating B-cell-like DLBCL (ABC-DLBCL). MYD88^L265P^-mutated ABC-DLBCL tumors with concomitant mutation in BCR signaling component CD79A/B responded to ibrutinib (80% response rate), but tumors harboring the MYD88^L265P^ mutation with wild-type CD79A/B were resistant to ibrutinib, suggesting that these tumors could probably use MYD88-dependent survival signaling [34]. Staudt et al. performed WES and transcriptome sequencing on 574 DLBCL tumors, which revealed four distinct genetic subtypes of DLBCL differing in their gene expression signature and response to chemo-immunotherapy. These genetically distinct subtypes in DLBCL included MCD (co-occurrence of MYD88^L265P^ and CD79B^mut^), BN2 (BCL6 fusions and NOTCH2 mutations), N1 (NOTCH1 mutations), and EZB (EZH2 mutations and BCL2 translocations). Among these groups, MCD and N1 were associated with inferior clinical outcomes compared to EZB and BN2 [21]. In a subsequent study, genomic characterization of the MCD group led to identification of inactivating mutations in *KLHL14* (subunits of Cullin-RING ubiquitin ligase, required for turnover of BCR subunits). These *KLHL14* mutations occurred frequently and were recently found to confer resistance to ibrutinib in ABD-DLBCL by promoting the assembly of MYD88-TLR9-BCR (My-T-BCR) supercomplex [22]. The My-T-BCR supercomplex has been previously attributed to ibrutinib-responsive subsets of ABC-DLBCL [35].

By WES, Chiron et al. first demonstrated BTK^C481S^ mutation as a mechanism of ibrutinib resistance in relapse MCL tumors, which was, however, absent in patients with primary ibrutinib resistance or in those who showed transient response to ibrutinib [36]. Rahal et al. demonstrated the genetic cause of primary ibrutinib resistance in MCL. Using ten MCL cell lines, (four sensitive and six resistant to ibrutinib), they found that sensitive cell lines display chronic activation of BCR signaling, whereas resistant lines were dependent on the MAP3K14-NF-κB pathway leading to NF-κB activation [37]. Further genomic studies have identified a loss of function mutation in NF-κB inhibitors (*TRAF2*, *TRAF3,* and *BIRC3*) associated with primary ibrutinib resistance in MCL cell lines. This observation was further confirmed in 165 primary MCL samples where 15% of the tumors that did not respond to ibrutinib treatment had mutations in *TRAF3* and *BIRC3* [37]. Importantly, mutation in *BIRC3* is associated with activation of non-canonical NF-κB signaling in MCL and CLL, which is quite different from classical NF-κB activation mediated through BCR signaling, and, therefore, this activation of non-canonical NF-κB signaling instructed resistance to ibrutinib [38,39]. Another report by Lenz et al. has confirmed the association of mutations in NF-κB inhibitors with primary ibrutinib resistance [40]. WES on 27 paired primary MCL samples from Chenglin et al. identified 18 recurrently mutated genes, including *ATM, MLL2, SIPR1*, and *CARD11*. These mutated genes were further screened in 173 additional MCL samples. Mutations in *CARD11*, which are present downstream of BTK and regulate NF-κB activity, were present in 5.5% (10/179) of MCL tumors and were associated with primary ibrutinib resistance [24]. Aggarwal et al. published data from a single-arm phase-2 clinical trial (NCT02471391) involving 24 R/R MCL patients who underwent ibrutinib treatment for four weeks followed by an add-on treatment with venetoclax (a BCL2 inhibitor). Out of the 24 patients, five did not respond to treatment. These five patients had either chromosome 9p loss containing the SMARCA2 genomic region (4/5), deletions in *ARID2* (3/5), or mutations in *SMARCA4* (3/5) [23]. In a recent study, Chr 9p, 6q and 13q have been identified as important in WES data from seven ibrutinib-resistant and seven ibrutinib-sensitive MCL tumors [25]. The origin of these genetic alterations could be from the selection of pre-existing mutant cells from the heterogeneous tumor population or de-novo genetic anomalies that ascend during tumor cell division and produce resistant phenotypes.

### 2.2. Non-Genetic Causes of Ibrutinib Resistance

Besides genetic aberrations, molecular changes have also been associated with the development of intrinsic and acquired ibrutinib resistance. Herein, we have discussed the intrinsic and non-genetic causes of ibrutinib resistance. Kim et al. revealed that short exposure (72 h) of ibrutinib at a lethal dose (IC90) to sensitive ABC-DLBCL cell lines led to the selection of intrinsically ibrutinib-resistant cells. These ibrutinib-resistant lines, after transcriptomic analysis, were found to overexpress CD79B and had Akt/MAPK activation [41]. Interestingly, no such mechanism of CD79B overexpression has been reported in ibrutinib-resistant CLL or MCL lines. By comparing the RNA profiles of ibrutinib-resistant and ibrutinib-sensitive MCL cell lines, Jimmy et al. identified MYC overexpression as the intrinsic cause of ibrutinib resistance [42]. Furthermore, they explored the application of a HSP90 inhibitor (PU-H71) as a therapeutic regimen to inhibit growth of primary ibrutinib-resistant MCL tumors in a patient-derived xenograft (PDX) mouse model [42]. Another report in MCL cell line has shown that the activation of non-canonical NF-κB and MAPK pathway through CD40L-CD40 signaling could bypass BTK signaling consequently led to decrease in ibrutinib efficacy [43,44,45]. XPO1, an exportin protein that regulates nuclear export of multiple oncogenic proteins is highly expressed in MCL tumors [46]. Targeting XPO1 with the selective inhibitor selinexor was found to sensitize intrinsic ibrutinib-resistant MCL cell lines by preventing the nuclear export of NF-κB regulators and inhibiting NF-kB signaling [46]. Detailed information concerning the bypass mechanisms associated with acquired non-genetic ibrutinib resistance development and their therapeutic strategies have been discussed in Section 4.

### 2.3. TME and Ibrutinib Resistance

TME has been shown to facilitate tumor cell growth through bidirectional interactions which occur either through direct contact between tumor cells and stromal cells or by indirect contact through cytokines and growth factors. Such bidirectional interactions have been known to contribute towards development of chemo-resistance. TME has been found to support tumor growth by activating beneficial signaling for tumor growth. For example, CLL cells in the lymph node have activated BCR signaling, whereas circulating CLL cells isolated from blood tend to be resting and in a quiescent state [47]. Mesenchymal stromal cells (MSCs) are multipotent, residing in various tissues and organs, and are a major constituent of stromal niches of TME. Once MSCs are activated, they secrete various cytokines and growth factors that modulate local immune responses and promote tumor cells growth [48,49]. Ibrutinib has considerable activity towards inhibition of signaling within TME. In fact, various cell types in the tumor environment express BTK, and these cells might be affected/suppressed by ibrutinib treatment. It is known that MSCs support the growth of DLBCL cell lines by up-regulating the CXCL12/CXCR4 axis. This oncogenic signaling axis has been found to be disrupted with ibrutinib monotherapy. Additionally, ibrutinib, when combined with a DNA-damaging agent, has shown evidence of synergistic killing activity [50]. The majority of studies integrating tumor cells with the microenvironment in lymphoma are in CLL. Bone-marrow-derived stromal cells have been found to support primary CLL growth by up-regulating mir-21. mir-21 expression was found to be inhibited by ibrutinib treatment [51]. MSCs isolated from CLL patients reinforced the growth of leukemic cells by secreting cytokines (IL-8, CCL4, CXCL10), a process that was found to be inhibited by ibrutinib [52]. Furthermore, ibrutinib induced the redistribution of CLL or MCL cells from the lymph node microenvironment to the peripheral blood. This was associated with changes in lymphoma cell surface markers which have known roles in cell activation and adhesion to stromal cells [9,53,54]. For example, CD19+CD5+ cells isolated from the peripheral blood of ibrutinib-treated MCL patients had a significant reduction in CXCR4, CD38 markers, and chemokine expression associated with adhesion and chemotaxis of MCL cells [10]. Another report showed that ibrutinib treatment significantly impaired integrin-α(4)β(1)-mediated adhesion of CLL cell lines or primary CLL cells to fibronectin or VCAM-1 [55]. Additionally, it was observed that ibrutinib treatment induced lymphocytosis in MCL and CLL patients (an increase in the number of circulating lymphocytes), suggesting that ibrutinib impairs integrin-mediated retention of malignant B-cells in the tumor-supporting lymph-node/bone marrow microenvironment.

Beside MSCs, myeloid cells are also important components of TME that are associated with cancer progression [56]. Wiestner et al. performed a study to understand ibrutinib-monotherapy-associated changes in tumor-specific macrophages (nurse-like cells; NLCs) using paired samples collected from a phase-2 clinical trial (NCT01500733) involving 80 CLL patients. In this study, a reduction in inflammatory cytokines in patient serum and bone marrow aspirate was observed along with a reduced number of circulating T-cells including Th17 T-cells. Most importantly, programmed-cell-death protein-1 (PD-1) expression on T-cells was significantly down-regulated [57]. Moreover, immunohistochemistry showed that ibrutinib treatment disrupted tumor-associated NLC–CLL interactions in the bone marrow [57]. Ibrutinib is also known to exhibit immunomodulatory properties by targeting multiple signaling pathways, as summarized in Figure 1. Kondo et al. showed the effect of ibrutinib on immune checkpoint protein expression [58]. In their study, peripheral blood collected from CLL patients treated with ibrutinib as monotherapy showed down-regulation of immune checkpoint PD-L1/PD1 expression on CLL/CD8+/CD4+ T-cells respectively, which was associated with STAT3 inhibition [58]. Ibrutinib treatment has also been found to increase in-vivo persistency of activated CD4+ and CD8+ T-cells, reduce Treg/CD4+ T-cell ratio, and downregulate immunosuppressive CD200 expression on CLL tumors [59]. A recent publication performed a multi-omics analysis (ATAC-sequencing, single-cell sequencing, and immuno-phenotyping) of the clinical time course (till 240 days of ibrutinib treatment) on CLL clinical samples that dissected the molecular responses to ibrutinib therapy [60]. They found time course acquisition of a quiescence-like gene signature in both CLL cells and non-malignant immune cells [60].

In AML, c-myc upregulation has been observed upon co-culture with mesenchymal stromal cells [61]. Moyo et al. showed that B-cell receptor signaling can be enhanced in precancerous B-cells, and cells overexpressing myc maintained a high degree of BCR signaling in spite of treatment with ibrutinib [62]. Currently, there is considerable interest in the arms race between ibrutinib and the TME, with the latter trying to enhance myc expression to resist ibrutinib pressure.

The role of ibrutinib is not limited to hematological malignancies, and there are a number of studies of solid tumors where ibrutinib has effectively regressed tumor progression and modulated TME. BTK inhibition by ibrutinib has led to prostate cancer cell apoptosis and dramatic changes in cell-adhesion-associated genes [63]. Ibrutinib treatment has also been found to improve the effectiveness of glioma therapy by modulating vascular permeability and preventing blood–brain barrier interference [64]. Ibrutinib has also shown positive responses with other inhibitors in solid tumors. Ibrutinib inhibited EGFR signaling in sorafenib-sensitive and sorafenib-resistant hepatocellular carcinoma cells (HCC) and showed synergistic effects with sorafenib in an immune-competent mouse model of HCC [65]. However, due to the off-target activity and non-specific interaction between ibrutinib and other kinases, the efficacy of ibrutinib has been limited.

Although these studies represent the efficiency of ibrutinib in modulating TME in a favorable direction, there are numerous contrasting reports as well. Zhao et al. demonstrated that TME supports the growth of MCL cells and can lead to ibrutinib resistance via activation of PI3K-mTOR pathway and integrin-β1 signaling [66]. Importantly, these changes were confirmed in both acquired (after chronic ibrutinib exposure) and de-novo (parental MCL cells co-cultured with stromal cells) ibrutinib-resistant MCL models [66]. Guan et al. found that co-culturing stromal cells with MCL cells imparted ibrutinib resistance via activation of the PI3K pathway and integrin-VLA-4 signaling. They also reported that blocking PI3K and integrin signaling sensitized MCL cells in TME condition [67]. Rudelius et al. identified enhanced focal adhesion kinase (FAK) activity in MCL in the MCL–stromal cell co-culture system and in primary MCL samples. These findings suggest that using a FAK inhibitor in conjunction with ibrutinib could potentially produce a synergistic therapeutic effect [68]. An in-vitro co-culture model of primary CLL cells with human bone marrow stromal cells (HS-5), maintained CLL cell proliferation and prevented ibrutinib-mediated cell killing irrespective of BTK mutation status [69]. Ibrutinib treatment has also been found to have a negative impact on the anti-tumor properties of NLCs. Ibrutinib-treated primary NLCs have reduced phagocytic ability, expressed immunosuppressive cytokines, and prevented ibrutinib-mediated primary CLL cell apoptosis [30]. Similar observations regarding the negative influence of ibrutinib therapy on NLCs in CLL have also been made by another group [70]. Although ibrutinib induces an egress of malignant lymphoma cells from their resident tissues, it does not induce the full egression of NLCs from resident niches, leaving a small fraction of these cells to interact with residual lymphoma cells and develop acquired resistance. Both the negative and positive impacts of ibrutinib therapy on TME modulation and conversely, TME effects on outcomes of ibrutinib therapy have been proposed, suggesting a complex multifactorial mechanism of action associated with ibrutinib.

### 2.4. Cancer Stem Cells (CSCs) and Ibrutinib Resistance

CSCs are subpopulations of cancer cells that have similar characteristics to normal stem cells or progenitor cells, such as self-renewal ability and the potential to differentiate. These properties of CSCs are responsible for driving tumor heterogeneity and developing drug resistance. MSCs have been shown to potentiate the growth of primary MCL-derived CSCs (CD133+CD45+CD19-) that grow in the form of cobblestones. MCL-derived CSCs possess stem-cell-like properties with a high expression of the stem-cell-specific genes *Oct4* and *Nanog*. Less than 100 of these cells can form complete heterogeneous tumors [71,72,73]. These cancer-initiating cells are resistant to genotoxic agents and ibrutinib, therefore requiring alternative targeting agents [74]. Although MSCs support the growth of CSCs, the resistance associated with ibrutinib therapy or other agents could be due to their intrinsic cellular properties. Besides MSCs, the functions of tumor-associated macrophages (TAMs) have been investigated in supporting and maintaining lymphoma CSCs through secretion of the growth factor pleiotrophin [75]. Studies on B-cell-lymphoma-specific CSCs are limited to mainly the MCL subtype, as the other subtypes (CLL, DLBCLs) are the clonal disorder of mature, differentiated lymphocytes. Previous reports have shown the existence of “side population” (SP) cells in lymphoma cell lines (DLBCL and Burkitt lymphoma), suggesting that these cells could be molecularly similar to CSCs [76]. SP cells extracted from a population of follicular lymphoma (FL) cells are highly enriched in CSCs. These SP cells are resistant to chemotherapy and radiation treatments [77]. Importantly, the growth of these FL-SP cells could be supported by tumor-associated follicular dendritic cells (FDC) through the CXCL12/CXCR4 chemokine signaling pathway [77]. CD45+CD19- is the defined marker of MCL-CSCs; however, no such defined marker has been identified for CSCs in DLBCL. A recent effort has been taken to isolate a CD45+CD19- population from DLBCL, but these cell populations did not recapitulate the molecular features of CSCs [78].

While ibrutinib is ineffective in targeting MCL-CSCs, it has shown promising activity in targeting CSCs in solid tumors. This effect could be due to an off-target activity of ibrutinib. Ibrutinib in combination with cisplatin has been shown to overcome cisplatin-resistance in ovarian cancer by diminishing the CSC population [79]. Malignant and stemness phenotype of glioblastoma, which has historically been associated with high BTK expression, could also be suppressed by ibrutinib treatment [80,81]. Ibrutinib has also been found to inhibit BMX (a non-receptor tyrosine kinase) mediate glioma stem cell (GSC)-derived pericyte growth and also disrupt the blood–tumor barrier, which augmented glioma-specific chemotherapeutic efficacy [82]. In multiple myeloma cells, enhanced BTK expression was implicated in increasing the features of cancer stemness that were previously inhibited by ibrutinib treatment [83].

## 3. Strategies to Overcome Ibrutinib Resistance

Understanding the genomic and molecular aberrations underlying ibrutinib resistance is one of the ways to identify likely therapeutic alternatives for R/R patients. Several studies have been performed or are currently underway to explore different strategies to overcome ibrutinib resistance. Some of these treatment strategies targeted towards ibrutinib resistance cases are discussed below and summarized in Figure 1.

### 3.1. New Generations of BTK Inhibitors

Ibrutinib, although selective towards its target BTK, has many off-targets, including interleukin-2-inducible T-cell kinase (ITK), epidermal growth factor receptor (EGFR), and phosphoinositide 3-kinase (PI3K) [84]. Adverse clinical effects of ibrutinib treatment including diarrhea (60%) and rashes (25%) in CLL patients that are often observed are associated with EGFR inhibition, and these side effects are also frequently observed in patients undergoing EGFR-targeted therapies [85]. A pooled analysis of four large, randomized, controlled studies which included 1505 CLL and MCL patients has shown an increased in incidence of atrial fibrillation in patients treated with ibrutinib therapy with frequency ranging from 4% to 10% above that expected in the general population [86]. Moreover, ibrutinib therapy is associated with a marked increase in the incidence of ventricular arrhythmias and sudden cardiac death [87,88]. The underlying mechanism of arrhythmias is not well documented but it could be associated with alleviation in cardiac PI3K/Akt signaling [89]. A multicentric cohort study noted hypertension as a common adverse event associated with patients who were treated with ibrutinib in a clinical trial setting, with a median time to reach peak blood pressure of 6 months (incidence rate of 18%; and for grade ≥3 hypertension in 6%) [90]. Increased risk of minor bleeding is an another ibrutinib-treatment-associated toxicity observed in up to 66% of patients, and this led to serious bleeding complications in patients who were on anticoagulants, such as warfarin or aspirin along with ibrutinib treatment [91]. This increased risk of bleeding in patients after ibrutinib treatment is attributed to on-target BTK inhibition, as patients with X-linked agammaglobulinemia (absence of BTK expression and defects in platelet aggregation) do not develop excessively bleeding deformities [92]. Arthralgia and myalgia of grade 1–2 have been reported during the initial course of ibrutinib treatment in nearly 20% of CLL patients, and more severe cases are rare. So far, there is no published data to confirm the causative mechanism of arthralgia due to ibrutinib treatment, but this could be because of off-target kinase activity of ibrutinib, as complications are less frequent with another BTK-specific inhibitor, acalabrutinib [93]. A pooled meta-analysis from seven randomized controlled studies of ibrutinib versus comparator included 2167 patients with CLL, MCL, and WM, who presented an increased risk of upper respiratory tract infection in the ibrutinib-treated group then in those treated with the comparator [94]. There are several mechanisms of increased risk of infection in ibrutinib-treated patients which have been reported. One such mechanism is impairments in immune cell function such as a reduction in natural killer cell antibody-dependent cellular cytotoxicity ability, and the phagocytic ability of macrophages [95,96]. Due to the off-target toxicities associated with ibrutinib treatment, nearly 30% of patients who enroll in ibrutinib therapy ultimately discontinue treatment. Second-generation BTK inhibitors are highly selective and are more potent than ibrutinib, including acalabrutinib (Acerta Pharma), zanubrutinib (BRUKINSA, BeiGene, Ltd., Beijing, China) and tirabrutinib (ONO/GS-4059, Ono Pharmaceutical, Osaka, Japan) [97]. In particular, acalabrutinib is gradually becoming more widely accepted than ibrutinib because of its limited toxicity [98]. Acalabrutinib has shown superior efficacy compared to ibrutinib in R/R CLL. This has led to its recent FDA approval as an initial or subsequent therapy for CLL. Approval of acalabrutinib was based on two clinical trials: ELEVATE-TN (NCT02475681; 535 CLL patients previously untreated) and ASCEND (NCT02970318; 310 R/R CLL patients) [99,100]. Similar to acalabrutinib, zanubrutinib has limited toxicity and was well tolerated in a phase-1 trial of CLL while producing a high response rate when used for patients with del17p [101,102]. Efficacy of zanubrutinib evaluated in a phase-2, BGB-3111-C206 (NCT03206970) clinical trial of 86 MCL patients has shown an ORR of 84%. A phase-1/2 trial, BGB-3111-AU-003 (NCT 02343120) consisting of 32 previously treated MCL patients, showed an ORR of 84%, leading its approval for treatment of MCL patients who have received at least one prior therapy. Table 2 represents recent FDA approvals along with respective clinical trials for ibrutinib, acalabrutinib and zanubrutinib. A recent publication from Zao et al. highlighted the effects of zanubrutinib treatment on immune cell re-modulation. Data from 25 R/R CLL/SLL showed that zanubrutinib treatment, like ibrutinib, led to reduction of PD-1 expression on CD4+ and CD8+ T-cells, PD-L1 on CLL cells, and inhibited suppressor cell functions [103].

### 3.2. Strategies to Target Secondary BTK^C481S^ or PLCG2^mut^

Ibrutinib covalently binds to BTK at the cysteine 481 position, irreversibly inactivating it, preventing downstream PLCG2 activation and BCR signaling [104]. The majority of reported ibrutinib-resistant cases are classified by a *BTK^C481S^* mutation, since ibrutinib is unable to bind irreversibly at cysteine 481 [19,105,106]. Some ibrutinib-resistant cases have reported mutations at the T316, T474, and L528 positions [14,31,32]. Although second-generation BTK inhibitors such as acalabrutinib, tirabrutinib, spebrutinib (CC-292) and BGB-3111 have demonstrated high clinical and preclinical efficacies [107], these inhibitors also bind irreversibly to BTK at cysteine 481 and, therefore, are not able to treat ibrutinib-resistant cases with a *BTK^C481S^* mutation. Novel third-generation BTK inhibitors and proteolysis-targeting chimeras (PROTACs) can effectively target BTK and mutant BTK^C481S^ and are currently in the early phase of clinical trials.

Mutations in the SH2 domain (R665W, S707Y, and L845F) of the BTK mediator PLCG2 keep PLCG2 in an active phosphorylated state even upon ibrutinib treatment, representing BTK-independent activation of a bypass pathway [17,18,108]. Strategies have been employed to target these mutant tumors with promising pre-clinical efficacies. Johnson et al. revealed a molecular mechanism conferring ibrutinib resistance through a PLCG2^R665W^ mutation in CLL [109]. Independent of BTK expression or mutation status, PLCG2^R665W^-mutant-expressing CLL cells confer hypermorphic induction of downstream signaling after BCR engagement. This study suggested targeting proximal kinases SYK or LYN could combat PLCG2^R665W^-dependent BCR signaling in ibrutinib-resistant cells [109]. Walliser et al. have shown that PLCG2^R665W^ and PLCG2^L485F^ mutants are hypersensitive to Rac2 (a Rho family of GTPase) activation, therefore, targeting Rac with specific inhibitors could provide a beneficial therapeutic regimen for PLCG-mutant-dependent ibrutinib-resistant cases [110]. Besides mutations in the SH2 domain of PLCG2 described here, several other PLCG2 mutations lying outside of the SH2 domain have been identified in ibrutinib-resistant cases; however, the effects of these mutations on PLCG2 activity have not yet been determined [111].

### 3.3. Third-Generation of BTK Inhibitors

ARQ-531 (ArQule Inc., Woburn, USA) is a potent, reversible, non-covalent inhibitor of BTK that occupies the ATP binding region within the kinase domain of BTK without interacting with C481. In a preclinical model, ARQ-531 significantly blocked BCR signaling in BTK^C481S^- and PLCG2-mutant-harboring CLL cell lines and patient samples [112]. A phase-1 dose-escalation study (NCT03162536) for ARQ-531 is currently underway in patients with multiple lymphoid malignancies to understand the safety, pharmacokinetics, pharmacodynamics, and clinical activity [113]. A recent publication has also demonstrated the in-vitro and in-vivo anti-proliferative activity of ARQ-531 in acute myeloid leukemia (AML) [114]. However, similar to ibrutinib, ARQ-531 has also shown non-specific activity on Src family kinases (Lyn) and kinases related to ERK signaling (MEK1) [112], therefore suggesting that ARQ-531 may have toxicity issues in clinical settings. LOXO-305 (Loxo Oncology, USA) is another reversible, non-covalent inhibitor for both wild-type and BTK^C481S^ mutant [115]. A phase-1/2 clinical trial (NCT03740529) for LOXO-305 is currently underway for R/R CLL patients who have failed or are intolerant to the standard of care. This trial will determine dose escalation and expansion while also establishing the maximum tolerated dose (MTD). Vecabrutinib (SNS-062, Sunesis Pharmaceuticals, USA) is a potent, reversible, non-covalent inhibitor of BTK, BTK^C481S,^ and ITK that has been shown to prevent auto-phosphorylation of BTK in whole blood from both humans and mice [116]. Currently, vecabrutinib is in a phase-1b clinical trial (NCT03037645) for multiple B-cell malignancies, including those who have failed prior standard of care therapies and patients with BTK^C481S^ mutations [117]. XMU-MP-3 is a recently identified non-covalent BTK inhibitor which also targets BTK^C481S^. XMU-MP-3 has been tested using MCL cell lines and showed promising efficacy in a MCL xenograft model [118]. Another drug, GDC-0853, has been tested for safety, tolerability, pharmacokinetics and activity in 24 R/R NHL or CLL patients, including in six patients with BTK^C481S^. Although well tolerated, an MTD for GDC-0853 could not be determined due to premature study closure [119]. The noncovalent BTK inhibitor, GNE-431, was assessed using x-ray crystallographic analysis for its binding and potency against BTK^C481S^ [120]. BTK inhibitors PRN1008 and BMS986142 have been tested in healthy volunteers and have been well tolerated with a sustained level of BTK occupancy [121,122]. Due to a lack of satisfactory information on GDC-0853, PRN1008, BMS986142, and GNE-431, further comments on these inhibitors cannot be made in this context.

### 3.4. BTK-PROTAC

Targeting undruggable proteins remains a serious challenge in drug development. Recent advancements have utilized the ubiquitin-proteasome system as a therapeutic approach for those disorders which are difficult to target with conventional inhibitor paradigms. One such approach is PROTAC, which brings and degrades a target of interest through its interaction with the E3 ligase (either Cereblon or Von Hippel–Lindau). Bromo-domain extra-terminal (BET) proteins targeting PROTAC (ARV-825 and ARV771; Arvinas, Inc., New Haven, CT, USA) have shown significant efficacy both in in-vitro and in-vivo xenograft models of Myc-dependent malignancies such as ABC-like DLBCL [123] and ibrutinib-resistant MCL [124]. Importantly, BET-PROTAC has been found to be more potent than a standalone BET inhibitor, as it could produce more perturbations in the mRNA and protein expressions and induce more apoptosis [124]. Due to the heightened potency of PROTAC for its target, the list of PROTAC-susceptible proteins is continually growing. Currently, RIPK2, ERRα, BRD4, BCR/Abl, several receptor tyrosine kinases, and histone modifier HDAC6 have been documented [125,126,127,128,129,130]. Therefore, PROTAC for BTK mutations could be a powerful tool to eradicate BTK-dependent malignancies. Huang et al. synthesized two distinct BTK degraders: (1) a bosutinib-based, cyanoquinoline BCR-ABL inhibitor; (2) an RN486-based aminopyridinone. Both of these BTK degraders have equivalent potency for targeting BTK. However, the RN486-based BTK degrader was more selective for BTK, whereas the bosutinib-based BTK degrader also inhibited Src and Tec family kinases [131]. Two different groups from different geographic regions (Crew group, USA; and Rao group, China) have designed and published research on BTK-PROTAC [132,133]. BTK-PROTAC, MT-802 (derived from an ibrutinib warhead lacking the meta-acrylamide moiety), developed by the Crew group, binds to fewer off-target kinases than ibrutinib and has equivalent potency for both wild-type and BTK^C481S^ [132]. Evaluation of in-vitro efficacies of MT-802 through treatment of CLL cells isolated from BTK^C481S^ patients showed a significant reduction in BTK phosphorylation [132]. Further advancements aimed at improving pharmacokinetic properties of MT-802 by E3-ligand and linker optimizations, which resulted in development of equally potent BTK-PROTAC “SJF620” [134]. BTK-degrader P13I generated by the Rao group has also been shown to induce significant degradation of wild-type and BTK^C481S^, with DC_50_ at 9.2 and 30 nM, respectively [133]. Unlike ibrutinib, the authors claimed that P13I (pomalidomide E3 ligase) has almost no inhibitory activity against other kinases such as ITK or EGFR, even at >1000nM inhibitory concentration [133]. In another follow-up article, the Rao group analyzed the potency and efficacy of P13I PROTAC in degrading BTK in mice. P13I is well tolerated in mice, with intraperitoneal administration of P13I at 33mg/kg three times daily for 11 days completely abrogating BTK expression in the adipose, thymus, lung and intestinal lymph nodes [135]. After withdrawal of P13I, BTK expression could be recovered within 7 days, suggesting that this PROTAC strategy functions as a reversible inhibitor for its target and could be utilized for therapeutic purposes [135]. In line with this, the Rao group developed a second-generation BTK-PROTAC “L18I” (lenalidomide E3 ligase) with improved aqueous solubility compared to the first-generation BTK degrader P13I [136]. This new BTK-PROTAC could degrade different BTK mutants with DC_50_ lower than 30nM. The new BTK-PROTAC induced rapid tumor regression of BTK-mutant-bearing lymphoma xenografts with limited signs of toxicity and showed even higher inhibitory activity when combined with Lyn, PI3K and Syk inhibitors [136]. This study has addressed multiple questions and opened the door for BTK-PROTAC to be clinically investigated for its efficacy and safety as a monotherapy or combination therapy for lymphoma. So far, BTK-PROTAC therapies are not in any current clinical trials. The efficacy of other gene-targeting PROTACs is under investigation in clinical trials. A list of BTK-PROTACs and their respective studies is in Table 3. Recently, Arivinas, a US-based biotechnology firm, has reported encouraging data from an ongoing phase-1 clinical trial for PROTAC, ARV-110 (which targets androgen receptors). A total of 36 participants with metastatic castration-resistant prostate cancer were involved and data from the first three cohorts showed that the drug was well tolerated without any dose-limiting toxicities (NCT03888612). Another currently ongoing clinical trial for Arvinas PROTAC is ARV-471 (estrogen receptor targeting) for patients with estrogen-positive/human epidermal growth factor receptor-2-negative (ER+/HER2-) advanced or metastatic breast cancer (NCT04072952). Based on the initial success of these described PROTACs in clinical trials and BTK-PROTAC efficiency in preclinical studies, it is worthwhile to investigate some of these BTK-PROTAC through clinical trials, either as a monotherapy or in combination therapies. A few limitations could potentially be associated with BTK-PROTAC studies in clinical trials: (1) non-specific toxicities may lead to discontinuation of the clinical trials; (2) BTK-PROTAC may not be able to provide beneficial results in ibrutinib-resistant cases such as those that are dependent on genetic alterations including mutant *PLCG2* or other genes; (3) BTK-PROTAC may not find success in eliminating BTK-independent ibrutinib-resistant tumors.

## 4. Acquired Ibrutinib-Resistance and Therapeutic Approaches

Ibrutinib, as a single agent, is effective in treating CLL, different subtypes of lymphoma and other B-cell malignancies unless unacceptable toxicity or disease progression is observed. Due to chronic exposure of ibrutinib during treatment, lymphoma cells could attain compensatory survival pathways, genetic mutations or clonal selection leading to the development of acquired or secondary resistance. In this section, deregulated pathways associated with acquired ibrutinib resistance have been discussed. Additionally, the therapeutic regimens targeting these alternative molecules have also been discussed.

### 4.1. PI3K/Akt/mTOR Pathway

Towards the goal of mimicking the clinical course of ibrutinib treatment, Wang et al. developed an ibrutinib-resistant MCL PDX mouse model through chronic ibrutinib exposure, that ultimately lead to the identification of several important survival pathways [137]. Among all other survival pathways, constitutive activation of PI3K/Akt/mTOR signaling is one of the commonly deregulated pathways identified in their acquired ibrutinib resistance PDX model. Inhibition of PI3K signaling in combination with ibrutinib reduced growth of ibrutinib-resistant tumors in the PDX model [137]. To mimic a similar condition, we and others have developed an in vitro acquired model of ibrutinib resistance for ACB-DLBCL cell lines by chronic exposure to ibrutinib. We have identified that up-regulation of PI3K/Akt/mTOR signaling can be targeted by selective PI3K isoform inhibitors to overcome ibrutinib resistance, either alone or in combination with the standard therapeutic regimen [138,139]. A similar strategy was followed by Kapoor et al. wherein they generated ibrutinib-resistant CLL and ABC-DLBCL lines by chronic exposure to ibrutinib and identified a PI3K/Akt signaling dependency [138]. Mutations in BCR signaling components including *CARD11, CD79A/B, TNFAIP3,* and *MYD88*, have been associated with constitutive activation of BCR signaling in ABC-DLBCL via activating NF-κB pathway and resistance to ibrutinib [140,141,142]. These BCR mutant bearing tumors are reliant on PI3K signaling to drive NF-κB activation and therefore, targeting the PI3K pathway using isoform-specific inhibitors has resulted in tumor regression in ibrutinib-resistant ABC-DLBCL in vivo models [143]. Although PI3K/Akt inhibition has shown strong antitumor activity in preclinical models, this inhibition was not profoundly effective in vivo in ibrutinib-resistant MCL-PDX models [66,137]. Transcriptomic profiling followed by NanoString nCounter gene expression analysis from two independent ibrutinib-resistant and sensitive cohorts (*n* = 13/23) identified significant enrichment of OXPHOS related genes in ibrutinib-resistant tumors, and treatment with OXPHOS inhibitor IACS-010759 could inhibit MCL-PDX tumor growth [25]. PLK1 kinase, which acts upstream of PI3K/Akt signaling via phosphorylating PTEN (a negative regulator of PI3K signaling), is found to be active in ibrutinib-resistant MCLs. Volasertib, a specific PLK1 inhibitor is currently in a phase-3 trial for patients with acute myeloid leukemia. Volasertib has been found to inhibit growth in an ibrutinib-venetoclax dual-resistant PDX model, suggesting that it may be another mode to target PI3K signaling [144].

### 4.2. Reduced BTK Expression after Chronic Ibrutinib Treatment

Down-regulation/loss of drug-target expression due to the clinical course of treatment is another approach exploited by tumor cells to develop acquired resistance. A study from Ghandi et al. observed a decline in total BTK expression in circulating CLL cells derived from the peripheral blood of ibrutinib-treated patients [145]. Similarly, we have also observed a significant reduction in total BTK expression in chronically exposed ABC-DLBCL lines [139]. The molecular mechanism underlying the reduction in total BTK expression after chronic ibrutinib exposure has not been investigated yet. This reduction in drug target expression could be due to the selection of those tumor cells from the heterogeneous population that have intrinsically reduced or null target gene expression before chemotherapy. Reduction in total BTK expression was not observed after minimal-time of ibrutinib exposure (72 h) [6], suggesting that prolonged ibrutinib exposure could potentially have selected lower BTK expressing cells, or could be due to ibrutinib associated epigenetic changes [11,146]. This is in concordance with down-regulation/loss of CD20 expression (B-cell membrane protein) after prolonged rituximab (monoclonal antibody for CD20) exposure that leads to development of resistance to rituximab [147]. It is yet to be determined in clinical studies whether the observed reduction in BTK expression is associated with chronic ibrutinib treatment leading to acquired ibrutinib resistance development.

### 4.3. BCL2 Signaling

BCL2 family proteins are central regulators of apoptosis and have recently emerged as a therapeutic target in B-cell lymphoma. Venetoclax (ABT-199), an oral first-in-class BCL2 inhibitor, showed antitumor activity in B-cell lymphoma [148]. Our data and others have revealed that chronic ibrutinib exposure can reprogram resistant cells to express high levels of BCL2 [139], and therefore venetoclax may have synergistic anti-proliferative activity when combined with ibrutinib for combination therapy in resistant ABC-DLBCL lines and primary tumors [149]. Similar results were obtained in the ibrutinib-resistant WM cell line model, acquired by sequentially escalating ibrutinib concentrations [150]. An ex vivo study from Deng et al. on primary CLL tumors proved that ibrutinib or acalabrutinib treatment reprogrammed lymphoma cells to BCL2 dependency and therefore, could be sensitized by combining ibrutinib with venetoclax [151]. Additionally, combinatorial drug screening with ibrutinib demonstrated the synergistic activity of BCL2 inhibition in MCLs [152]. Synergetic behavior of combining ibrutinib and venetoclax has already been established in a recent phase-2 clinical trial (NCT02756897) of 80 CLL patients (having genetic abnormalities; del17p or un-mutated IGHV), that included patients who received ibrutinib monotherapy for the first three cycles followed by venetoclax for 12 cycles. The treatment was well-tolerated, and nearly 61% of patients achieved complete remission with undetectable or minimal residual disease [153]. A recent publication from Tyner et al. concluded after assessing 651 primary cells through an ex vivo functional screening that CLL and AML patients had a significantly higher sensitivity to ibrutinib + venetoclax combination therapy compared to the individual treatments alone [154]. Although these studies suggest that combining BCL2 inhibition with BTK inhibition could strengthen the therapeutic efficacy in ibrutinib resistance patients, this strategy needs further investigation in subsequent clinical trials to eliminate possibilities of unexpected heterogeneous de novo resistance to these combinations [155].

### 4.4. Bromodomain and Extraterminal Domain-Containing Proteins (BETs) Inhibitors

BET proteins such as BRD4 are positioned preferentially at the hyper-acetylated super-enhancer regions in chromatin via its two bromodomains and regulates expression of important oncogenes including *Myc, BCL6, BCL-XL*, and *TCF4* [123,125]. Although BET-inhibitor JQ1 have been shown to attenuate *MYC, CDK4/6, RelA, BTK* and *NF-κB* target gene expression in MCL cells, could synergistically induce apoptosis of ibrutinib-resistant MCL cells and has also shown some synergistic activity, mediated by NF-kB pathway blockade, in ABC-DLBCL [156,157], this inhibitor also robustly led to BRD4 protein accumulation, which has been implicated in inadequate MYC suppression [125]. Compared to BET-inhibitors (JQ1 or OTX015), BET-PROTACs treatment (ARV-771 or ARV-825) induced more perturbations in gene expression, and have shown promising anti-tumor activity in multiple malignancies including DLBCL [123,125]. Importantly, ibrutinib-resistant MCL cells that are sensitive to ARV-771 treatment have showed synergetic activity with ibrutinib both in vitro and in vivo conditions [124]. Recently, a novel BET-inhibitor, GS-5829, has been pre-clinically tested in CLL, which inhibited key signaling pathways and several oncogenes including *MYC, Akt, ERK1/2, NF-κB,* and *BLK*. GS-5829 induced apoptosis of primary CLL cells irrespective of their IGHV mutational status [158]. Moreover, GS-5829 also has shown evidence of a synergetic combination with ibrutinib [158]. Although GS-5829 can significantly downregulate the expression of multiple signaling proteins, its effectiveness in reducing BRD4 expression or other BED4 dependent signaling proteins (Myc) at equimolar concentrations has not been compared with BET-PROTAC.

### 4.5. MALT1 Inhibition

MALT1 is an essential component of the CARD11-BCL10-MALT1 (CBM) signaling complex that links BCR signaling to the NF-κB pathway. MALT1 activity is up-regulated in lymphoid malignancies such as ABC-DLBCL. As such, MALT1 inhibition can potentially selectively inhibit the growth of ABC-DLBCL PDX tumors [159]. Given the prospective anti-tumorigenic role of MALT1 inhibitor (MI-2), Saba et al. recently investigated ex-vivo efficacy of MI-2 treatment in primary CLL cells including ibrutinib-resistant CLL tumors. MI-2 treatment significantly inhibited ibrutinib-resistant primary CLL cell growth in a dose-dependent manner [160]. Another report from Dai et al. also implicated MALT1 inhibition as a potential strategy to overcome secondary ibrutinib resistance caused by BTK mutations in MCL cells [161].

### 4.6. IRAK4 Inhibition

Toll-like receptor signaling (TLR) adapter protein MYD88^L256P^ is frequently mutated (29%) in ABC-DLBCL, and accountable for IRAK4 kinase-mediated activation of NF-κB and TLR signaling. Inhibition of IRAK4 with a specific inhibitor in MYD88^L256P^ mutant ABC-DLBCLs could prevent NF-κB signaling, and inhibited tumor growth when combined with ibrutinib [162]. Vincenza et al. presented similar in vitro data with IRAK4 inhibitor in CLL cells [163]. Additionally, matrix high-throughput drug combination screening has revealed a synergistic relationship of ibrutinib with a bromodomain inhibitor (JQ1), XPO1 inhibitor (selinexor), and IRAK4 inhibitor in ABC-DLBCL cell lines [164]. Current research on IRAK4 has been focused on developing pharmacologically efficacious IRAK4 inhibitors for therapeutic use [165,166].

### 4.7. SYK Inhibition

SYK kinase is an important upstream player of BTK activation in BCR signaling, and has shown to be up-regulated at both the RNA and protein level in CLL [167]. Besides antigen-dependent BCR stimulation, SYK activity has also been shown to be regulated by mutated MYD88^L256P^ that forms the myddosome signaling complex and activates STAT3 and Akt signaling in MYD88^L256P^ lymphoma cells [168]. MYD88^L256P^ mutation is associated with ibrutinib resistance and is frequently found in ABC-DLBCL (30%) and WM (90%). As such, SYK has the potential to be an excellent therapeutic candidate. A recent study from Munshi et al. reported that combined SYK (tamatinib) and BTK (ibrutinib) inhibition could induce synthetic lethality in MYD88^L256P^ lymphoma cells [168]. Importantly, SYK inhibition also abrogated survival signals from TME and in combination with ibrutinib synergistically induced apoptosis in primary CLL cells [169]. JAK-STAT is an important pathogenic signaling pathway in CLL which through cytokine secretion can mediate survival in the backdrop of CLL-TME interaction [69]. Wang et al. explored the therapeutic efficiency of cerdulatinib (a novel orally available dual JAK/SYK inhibitor) in primary CLL cells [69]. Treatment with cerdulatinib blocked JAK-STAT, BCR, and NF-κB signaling and induced primary CLL cell apoptosis in a CLL-HS-5 co-culture system [69]. Importantly, cerdulatinib treatment blocked the proliferation of BTK^C481S^ bearing ibrutinib-resistant primary CLL cells that were isolated from ibrutinib relapsed patients [69].

### 4.8. Chromatin Modifiers

Epigenetic aberrations are quite common in B-cell malignancies that display genome-wide DNA modifications such as acetylation, de-acetylation, hypo-methylation, and hyper-methylation at specific regulatory elements. These epigenetic modifications are directed by chromatin-modifying enzymes including histone acetyl-transferase (HATs), histone deacetylase (HDACs), and DNA methyl-transferase (Dnmts), resulting in changes of gene expression [170,171]. Although, ibrutinib has been widely studied for its role in targeting BCR signaling via BTK a few reports have also demonstrated a role of ibrutinib in regulating epigenetic modifications in B-cell lymphoma [11]. The transcription factor *NFATC1,* which is a downstream effector of BCR signaling, is hypo-methylated and its expression levels along with expression levels of its target genes (*BCL2, CCND1*) have been can be directly correlated to CLL progression This hypo-methylation and expression of NFATC1 could be potentially blocked by ibrutinib in CLL treatment [172]. A recent study has applied a systematic approach to identify the gene-regulatory landscape associated with ibrutinib treatment using 18 matched PBMCs from relapsed CLL patients, collected before or during ibrutinib treatment. In this study, ATAC-sequencing on matched tumors identified significant changes in chromatin accessibility in 616 regulatory elements (92% with lost and 8% with gain chromatin accessibility after ibrutinib treatment). Multiple genomic regions that had accessibility before treatment were lost during ibrutinib treatment and vice versa [11]. Further integrating this chromatin profiling with single-cell chemo-sensitivity profiling for 131 promising drugs, the authors identified ibrutinib-induced pharmacologically exploitable vulnerabilities such as proteasome inhibitors, PLK1 inhibitors, and mTOR inhibitors [11]. Another recent article investigated global changes in histone markers that were associated with ibrutinib treatment. Loss of both H3K27ac and H3K27me3 markers were identified after ibrutinib treatment compared to treated naïve CLL tumors and this was associated with changes in EZH2 gene expression [146].

Various groups have reported elevated expression of chromatin modifiers HDACs particularly HDAC6 in B-cell lymphomas compared to normal B-cells which can be directly correlated with disease progression. Inhibition of HDAC6 with selective inhibitor provided survival benefits to CLL in euTCL1 mouse model [173,174]. Importantly, inhibition of HDAC and PI3K signaling together with CUDC-907, a dual HDAC-PI3K inhibitor showed profound activity in ibrutinib-resistant MCL cell lines and PDX mouse models [175]. It is quite well-known that ABC-like DLBCL are sensitive to ibrutinib whereas germinal center-like (GCB) DLBCL are largely resistant [34]. MYD88^L256P^ mutation can impart ibrutinib resistance in ABC-DLBCLs in which expression of MYD88^L256P^ can be transcriptionally controlled by treatment with HDAC inhibitor panobinostat. Treatment with panobinostat with ibrutinib synergistically sensitized MYD88 mutant ABC-DLBCL cell lines to ibrutinib [176]. Recently, similar synergistic activity of HDAC6 inhibitor (ACY-1215) with ibrutinib was observed in FL cells [177].

EZH2 is another chromatin modifier (histone methyltransferase) whose activating mutations are enriched in germinal center-like DLBCLs [178]. EZH2 inhibitor (tazemetostat) has shown promising synergistic activity with ibrutinib in both EZH2 wild-type and mutant DLBCLs [178]. Other epigenetic modifier targeting drugs include PRMT5, a type II protein arginine methyl-transferase that catalyzes dimethylation of arginine residues on histone tails (H3R8 and H4R3) and other proteins, driving expression and activity of key oncogenes including cyclin D1, c-MYC, and NOTCH1 while also promoting lymphomagenesis [179]. A recent report identified that PRMT5 is overexpressed in MCL, and application of PRMT5 specific inhibitor PRT382 was found to reduce tumor burden in an ibrutinib-resistant MCL-PDX mouse model [180].

### 4.9. Ibrutinib in Combination with CD20 Targeting Immunotherapy

Rituximab, the first monoclonal antibody targeting the surface receptor “CD20” has been implemented in the therapy of B-cell lymphomas and managed to produce significant improvements in therapeutic outcomes. Obinutuzumab is a 2nd generation anti-CD20 monoclonal antibody which has higher antibody-dependent cellular cytotoxicity compared to rituximab. Multiple clinical trials looking to evaluate the therapeutic efficacy for ibrutinib in combination with an immunotherapeutic such as anti-CD20 monoclonal antibodies has either been completed or currently underway. Successful therapeutic response was reported in a case study, where a 73-year-old ibrutinib-resistant MCL patient achieved complete remission after three cycles of combination treatment with obinutuzumab and venetoclax [181]. In a phase-3 clinical trial for WM patients, the combination of ibrutinib with rituximab resulted in significantly higher rates of 30 month PFS (82%) compared to rituximab alone (28%) [182]. A recently completed multicenter randomized open-label phase-3 clinical trial-ILLUMINATE (NCT02264574) compared results from two combination treatment groups: Ibrutinib plus obinutuzumab (*n* = 113, CLL) and chlorambucil plus obinutuzumab (*n* = 116). 30-month progression-free survival on ibrutinib plus obinutuzumab group was reported to be significantly higher (79%) compared to the chlorambucil plus obinutuzumab group (31%) [183]. However, not all clinical studies have yielded similar results. A clinical study on CLL patients (irrespective of 17p deletion or TP53 mutation), identified no such clinical improvement in outcome between ibrutinib (*n* = 182, PFS 86%) and ibrutinib plus rituximab combination (*n* = 182, PFS 86.9%) treated groups [184,185] suggesting that addition of rituximab to ibrutinib was not associated with synergistic benefits to CLL therapy. Skarzynski et al. have reported that ibrutinib treatment can result in a rapid and persistent down-regulation of CD20 expression (74% reduction at 28 days of ibrutinib treatment, compared to baseline sample) on primary CLL tumors which could potentially explain the lack of enhanced therapeutic efficacy of rituximab-ibrutinib combination therapy when compared to ibrutinib monotherapy in CLL (NCT01500733) [186]. Most likely, ibrutinib-rituximab combination therapy is preferentially killing CD20 positive CLL cells from the heterogeneous tumor pool, leaving unaltered or clonally selected CD20 negative rituximab resistant CLL cells. This negative impact of ibrutinib therapy on CD20 expression can also be corrected with chromatin modifier HDAC inhibitor treatment as discussed previously [187]. Therefore, the addition of such agents with rituximab therapy can potentially improve ibrutinib efficacy and should be used clinically to evaluate treatment response in lymphoma patients, particularly in CLL. A list of clinical trials involving rituximab or its combination with ibrutinib or other agents for treatment of ibrutinib resistance cases has been provided in Table 4.

## 5. Therapies Targeting Ibrutinib-Resistant CSCs

Most therapeutic strategies are typically directed at the fast-growing tumor mass but not to the slow-dividing CSCs, implying that CSCs may survive therapeutic interventions due to their high resistance to drugs and slower proliferation rate. CSCs and hematopoietic stem cells share the Wnt, Notch, and Hedgehog signaling pathways, which are required for their growth and self-renewal. Therefore, it is important to develop CSC-specific therapies that avoid potential significant side effects caused by inhibition of normal stem cell functions. Thus far, very little is known about B-cell-lymphoma-associated CSC biology and a therapeutic regimen that may be effective. Mathur et al. have demonstrated that MCL-derived CSCs are resistant to ibrutinib and have up-regulated Wnt signaling. Targeting these MCL-CSCs by Wnt pathway inhibitors could preferentially eliminate these cell populations [74]. Other studies suggest that BMI-1 (a proto-oncogene) or reactive oxygen species (ROS) are required for MCL-CSC self-renewal and maintenance and, therefore, CSCs can be targeted by BMI-1- or ROS-directed therapy [194,195,196].

## 6. Chimeric Antigen Receptor (CAR) T-Cell Therapy with Ibrutinib for Combination Treatment

CAR T-cell therapy is an adaptive form of T-cell therapy, in which T-cells are genetically engineered to express an artificial receptor that targets a tumor cell surface antigen. CD19-directed CAR T-cell therapy has shown promising results in patients with R/R B-cell lymphoma in multiple clinical trials: ZUMA trials with Axicabtagene ciloleucel (NCT02348216, NCT02601313); the JULIET trial with Tisagenlecleucel (NCT02445248); and the TRANSCEND trial with Lisocabtagene Maraleucel (NCT02631044) [197]. Despite initial promising results in terms of therapeutic efficacy, certain limitations, such as unavailability of robust CAR T-cell expansion protocols, non-specific or failed engraftments, and associated toxicities (cytokine release syndrome, CRS) have been obstacles towards the integration of this unique therapeutic strategy into standard treatment regimens [198] Interestingly, it has been reported that these obstacles can be overcome by prolonged ibrutinib treatments (≥5 cycles) before autologous T-cell collection for CAR manipulation. This protocol could potentially aid in improving infused CAR T-cell expansion while reducing immunosuppressive marker “PD-1” expression on CAR T-cells and CD200 expression on B-lymphoma cells, respectively [199,200]. Concurrent ibrutinib and CAR T-cell therapy has been well tolerated, with no signs of impairment in CAR T-cell function. It has shown improved CAR T-cell engraftment and has been highly effective as a therapeutic regimen for R/R CLL patients (NCT01747486, NCT01105247, and NCT01217749) [199]. A recent study of high-risk CLL patients (19 patients, 17p deletion and/or complex karyotype) showed a 4-week ORR of 83% and a significant reduction in CRS when concurrently treated with CAR T therapy and ibrutinib [12]. A recent case report has shown data from a del(17p) CLL patient who developed resistance to BCL-2 inhibitor (Venetoclax) treatment with rapid disease progression within 3 months of treatment initiation. This patient was subsequently treated with combined ibrutinib and CAR T therapy and went on to achieve complete remission with no detectable minimal residual disease in the bone marrow and peripheral blood, within one month of treatment [192]. CAR T therapy has also shown durable molecular remission (4-week ORR of 71%) in CLL patients who were resistant to ibrutinib before CAR T therapy initiation [201]. Similar results were obtained in an MCL in-vivo model, where long term remission for CAR T-cell + ibrutinib treatment was 80–100%, compared to 0–20% for a CAR T-cell therapy only treatment group [202]. Since ibrutinib has off-target toxicity, second-generation BTK inhibitors such as acalabrutinib have also been tested for efficacy in combination with CAR T therapy. Similar to ibrutinib, acalabrutinib in combination with CAR T therapy improved CD19 tumor clearance and prolonged mice survival in vivo. Furthermore, acalabrutinib + CAR-T combination therapy could also potentiate CAR T therapy responses in B-cell malignancies in clinical trials [203] Currently, two big phase-1/2 clinical trials of JCAR017 as monotherapy or in combination with ibrutinib or other agents (cyclophosphamide/fludarabine) are ongoing (NCT03331198; 200 CLL/SLL), (NCT02631044; 274 multiple B-cell malignancies). Initial data from a limited number of patients has shown good safety profiles with improved anti-tumor responses. A list of clinical trials associated with CAR T therapies with or without combinations of other drugs is represented in Table 4.

## 7. Conclusions and Future Directions

Development of chemotherapeutic resistance is currently one of the main challenges plaguing anti-cancer therapy and is often responsible for cancer progression and treatment failure despite robust therapeutic interventions. In the presence of therapeutic stress, both the tumor and its microenvironment cells produce a series of responses, which include initiation of several complex feedback loops that combine to produce a survival network that contributes towards acquired resistance development. The development of ibrutinib resistance does not rely on a single aberration to lymphoma cells, but numerous genetic, non-genetic, primary, acquired, and other undiscovered mechanisms. Identification and understating of these aberrant molecular drivers could help in predicting disease progression and aid in developing new therapies for R/R ibrutinib resistance cases. So far, a variety of genomic mutations on BTK, PLCG2, and other important tumor suppressors/oncogenes have been identified as associated with ibrutinib resistance. Moreover, BTK-independent pathways are also reported to be prime players involved in ibrutinib resistance development. Identification of a common pathogenic driver from these vast molecular assaults causing ibrutinib resistance development is therefore a critical first step. In this regard, several new therapeutic targets and their respective inhibitors have been identified. For example, PI3K pathway targeting agents, BCL2 inhibitors, and HDAC inhibitors have been found to be promising in combination with ibrutinib as ibrutinib-failure therapies. Novel BTK inhibitors and specific PROTACs can also potentially function alone or in combination with other agents to eradicate BTK-mutant resistant tumors. The use of CAR-T therapy in combination with ibrutinib or other agents (venetoclax) can be another treatment-transformative approach. Results from an increasing number of pre-clinical and clinical studies may further guide to us towards successfully identifying treatment options. Due to the dynamics and complexities of tumor evolution, investigators and clinicians are required to observe drug-induced changes during treatment in a timely manner. A high-throughput drug screening platform can be of huge benefit to identify novel drug combinations for each person and will eventually help in developing precision medicine.

## Figures and Tables

**Figure 1 cancers-12-01328-f001:**
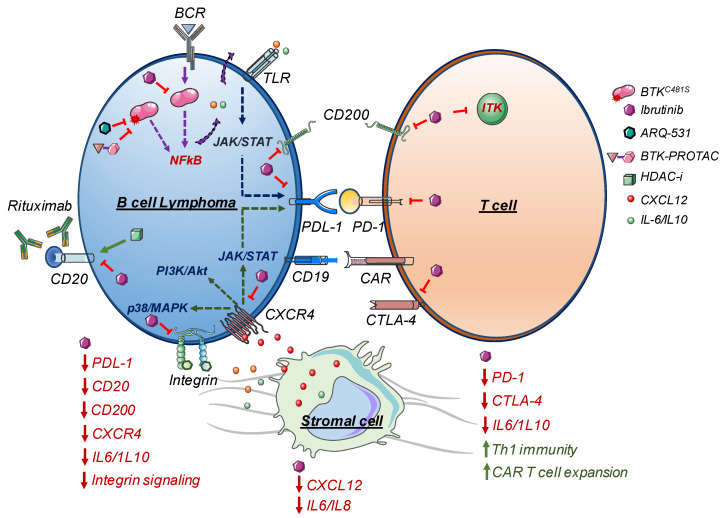
Ibrutinib’s mechanism of action and strategies to overcome ibrutinib resistance. Ibrutinib treatment has been shown to negatively influence TME and have immunomodulatory functions. Ibrutinib inhibits JAK/STAT signaling through CXCR4/CXCL12, thereby preventing expression of immunosuppression of PD-L1/CD200 on tumor cells and PD1/CTLA4 on T-cells. Additionally, ibrutinib has shown to promote Th1-type immunity and, therefore, significantly improved CAR T-cell expansion upon ibrutinib treatment. Ibrutinib is unable to target BTK mutants; therefore, third-generation BTK inhibitors or BTK-PROTAC can prevent BTK-mutant-dependent ibrutinib resistance development.

**Table 1 cancers-12-01328-t001:** Selected next-generation-sequencing-based studies that identified alternative genetic aberrations other than *BTK* or *PLCG2* mutations, acquired or clonally selected during disease progression to ibrutinib resistance.

Study	Method	Major Findings	Size	Reference
DLBCL	WES and transcriptomics	Inactivating mutation of *KLHL14*, enriched in 29.6% tumors of MYD88^L265P^-CD79B subtype	574 (biopsy)	Schmitz [21] Jaewoo [22]
MCL	WES, and TDS	9p21.1–p24.3 loss and/or mutations in components of SWI–SNF chromatin-remodelling complex	24 (R/R)	Aggarwal [23]
	WES on IS and IR	CARD11 mutation in 5.5% of cases	13	Chenglin [24]
	WES on 7 IS, 7 IR	Changes in DNA copy number alteration, broad deletions of 6q, 9p, and chromosome 13	37	Zhang [25]
FL	TDS panel of 140 genes, on pre-ibrutinib treatment	CARD11 (16%) and predicted resistance to ibrutinib (NCT01849263)	31 (biopsy)	Bartlett [26]
WM	WES on ibrutinib progressed tumors	Homozygous loss of chr; 6q and 8q at baseline (33% and 66%), at progression (60% and 80%) in tumor of MYD88^L265P^	5 (biopsy)	Jimenez [27]
	AS-PCR for MYD88 and CXCR4 mutation followed by ibrutinib response	Major response rate; MYD88^L265P^CXCR4^WT^ (91.2%), MYD88^L265P^CXCR4^WHIM^ (61.9%); Clinical Trial (NCT01614821)	63 (biopsy)	Treon [28]
CLL	TDS, for mutations in 29 genes	Mutation in TP53, SF3B1, and CARD11 genes	11 (paired)	Shamanna [29]
	WES and SNP 6.0 array profiling	Acquired or increased status of del17p/TP53 mutation in three out of five ibrutinib-resistant cases.	48 (paired)	Amin [13]
	WES and TDS	Chr;8p del with additional driver mutations (EP300, MLL2 and EIF2A)	5	Burger [30]
	A hybrid capture or SNV for panel with 1200 or 1212 CAG	BTK^T316A^ mutation confer activation of PLCG2	1 and 9	Sharma [31], Kadri [32].

Abbreviations: WES, whole exome sequencing; TDS, targeted deep sequencing; SNV, single-nucleotide variations; CLL, chronic lymphocytic leukemia; WM, Waldenstrom macroglobulinemia; MCL mantle cell lymphoma; DLBCL, diffuse large B-cell lymphoma; IS, ibrutinib-sensitive; IR, ibrutinib-resistant; R/R, relapsed/refractory; CAG, cancer-associated genes; AS-PCR, allele-specific polymerase chain reaction.

**Table 2 cancers-12-01328-t002:** Summary of FDA-approved BTK first- and second-generation inhibitors.

Compound	FDA Approved	Date of FDA	Study	Outcome	Adverse Events	CT Identifier
Ibrutinib (Imbruvica); Janssen Biotech, Inc	With Obinutuzumab for TN CLL	28 Jan,2019	Phase-3, TN 229 CLL	30-month PFS 79% (95% CI 70–85).	Grade 3–4, neutropenia, thrombocytopenia	NCT02264574 (PCYC-1130)
	With Rituximab for WM	27 Aug,2018	Phase-3, 150 WM	30-month PFS 82% (HR,0.20; P < 0.001)	diarrhea, arthralgia; AF: (12% with ibrutinib vs. 1% with Rituximab)	NCT02165397 (PCYC-1127)
	GVDH	2 Aug,2017	Phase-2, 42 GVHD	13.9-month ORR 67%	fatigue, diarrhea, muscle spasms, nausea, bruising	NCT02195869 (PCYC-1129-CA)
Acalabrutinib (ACP-196, (Calquence); AstraZeneca	With Obinutuzumab or monotherapy	21 Nov,2019	Phase-3, 535 TN CLL	HR, 95% CI, 0.006–0.17; P < 0.0001)	neutropenia, 31% vs. 11%, in acalabrutinib plus obinutuzumab vs. acalabrutinib	ELEVATE-TN (ACE-CL-007)
			Phase-3, 306 R/R CLL	HR, 0.31; 95% CI, 0.20–0.49; P < 0.0001	Calquence vs. other group AF: 5% vs. 3%; bleeding: 26% vs. 8%.	ASCEND (ACE-CL-309)
	R/R MCL, one prior therapy	31 Oct,2017	Phase-2, 124 MCL	15·2-months ORR 80%	Grade 1–2 myalgia (21%), diarrhea (31%); Grade 3–4 neutropenia (10%)	NCT02213926 (ACE-LY-004)
Zanubrutinib (Brukinsa); BeiGene Ltd.	MCL, one prior therapy	14 Nov,2019	Phase-2, 86 R/R MCL	18.4-month ORR 84%	Any grade, neutropenia (31.4%), URTI (29.1%), rash (29.1%).	NCT03206970 (BGB-3111-206)
			Phase-1/2, TN 32 MCL	18.8-month ORR 84%	Grade ≥ 3 (≥5%) were neutropenia, pneumonia, thrombocytopenia, and leukopenia	NCT02343120 (BGB-3111-AU-003)

Abbreviations: CLL, chronic lymphocytic leukemia; WM, Waldenstrom macroglobulinemia; MCL mantle cell lymphoma; GVDH, graft-versus-host disease; R/R, relapsed/refractory; ORR, overall response rate; PFS, progression-free survival; TN, treatment-naïve; HR, hazard ratio; AF, atrial fibrillation; URTI, upper respiratory tract infection.

**Table 3 cancers-12-01328-t003:** List of BTK-PROTAC currently under investigation.

BTK-PROTAC	Potency/Efficacy	E3 protein-ligand	PROTAC-Structure	Reference
MT-802	>99% degradation at 250 nM conc., more potent than ibrutinib, not suitable for in-vivo studies.	CRBN (C5)	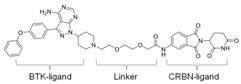	Buhimschi [132]
SJF620	Equivalent potency to MT-802, can be used for in-vivo studies.	CRBN (lenalidomide analog)	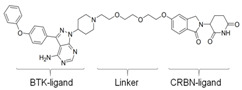	Figueroa [134]
P13I	89% BTK degradation at 100nM.	CRBN (pomalidomide)	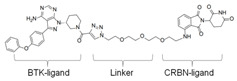	Sun [133]
L18I	Improved solubility vs. P13I in PBS	CRBN (lenalidomide)	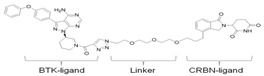	Sun [136]
CJH-005-067	Efficient degradation of BTK at 100 nM conc., bosutinib-based	CRBN (pomalidomide)	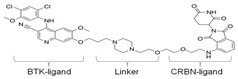	Huang [131]
DD-04-015	Efficient degradation of BTK at 100 nM conc., RN486-based	CRBN (pomalidomide)	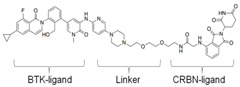	Huang [131]

Abbreviations: CRBN, cereblon; WT, wild type; PBS, phosphate buffer saline; CONC, concentration; nM, nano-molar.

**Table 4 cancers-12-01328-t004:** Summary of clinical trials representing immunotherapeutic agents with ibrutinib combinations.

Study	Patient	Size (M-Age)	Regimen	Outcome	Adverse Events	CT Identifier
Phase-3; Moreno [188]	CLL	229 (65)	Obinutuzumab-IB vs. obinutuzumab- chlorambucil	30-month PFS 79% vs. 31%	Serious adverse events: 58% vs. 35%	NCT02264574 (iLLUMINATE)
Phase-2; Burger [185]	CLL	208 (65)	Rituximab-IB vs. IB	36-month PFS 86.9% vs. 86%	Grade 3/4 TEAE: 65% vs. 64%	NCT02007044
Phase-3; Woyach [184]	CLL	547 (≥65)	BM-Rituximab vs IB-Rituximab vs. IB	38-month OS –not significant	Grade 3-5 hematological adverse events: 61% vs. 39% vs. 41%	NCT01886872
Phase-3; Meletios [182]	WM	150 (69)	Rituximab-IB vs. Rituximab	30-month PFS 82% vs. 28%	AF: (12% vs. 1%) Hypertension: (13% vs. 4%)	NCT02165397
Phase-3; Meletios [189]	WM (Rituximab refractory)	31 (67)	IB	18-month PFS 86% OS 97%	Grade-3 events: Neutropenia-13%, Hypertension-10%.	NCT02165397 (iNNOVATE)
Phase-3; Khan [190]	CLL	578 (≥18)	BM-rituximab vs. BM-Rituximab-IB	18-month PFS 24% vs. 79%	Grade 3-4 neutropenia: 51% vs. 54%	NCT01611090 (HELIOS)
Phase-2; Wang [191]	MCL (R/R)	50 (67)	Rituximab-IB	16.5-month PR 44% CR 44%	Grade 3 AF: 12% Grade 4 neutropenia: 1 patient	NCT01880567
Gauthier [12]	CLL (R/R) with del (17p)	19	CD19 CAR-T-cell-IB	4-week ORR 83%	Lower CRS after addition of IB	-
Gong [192]	CLL R/R to venetoclax, del(17p)	1	CD19 CAR-T-cell-IB	1-month CR, negligible MRD	Grade 1 CRS	-
Phase-1/2; Gauthier [193]	CLL (IB-resistant)	43	JCAR014-Cy-Flu-IB vs. JCAR014-Cy-Flu	4-week ORR 88% vs. 56%	No difference in grade ≥3 cytopenias, similar grade ≥1 CRS	NCT01865617
Phase-1/2; multicenter study	CLL/SLL	200	JCAR017 or JCAR017-IB	on-going	-	NCT03331198
Phase-1	Multiple B-cell malignancies	274	JCAR017	on-going	-	NCT02631044

Abbreviations: IB, ibrutinib; BM, bendamustine; PFS, progression-free survival; OS, overall survival; CR, complete response; PR, partial response; CLL, chronic lymphocytic leukemia; MCL, mantle cell lymphoma; WM, Waldenström’s macroglobulinemia; CT, clinical trial; M-age, median-age; Cy-Flu, cyclophosphamide, and fludarabine; AF, atrial fibrillation; TEAE, treatment emergence adverse event; CRS, cytokine release syndrome.

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
