# Peer review of "Ibrutinib Resistance Mechanisms and Treatment Strategies for B-Cell Lymphomas"

_cancers, 2020, doi:10.3390/cancers12051328_

Round 1
Reviewer 1 Report
In this manuscript, George et al review the recent advances made in the knowledge of the mechanisms of resistance to the first-in-class Bruton’s tyrosine kinase (BTK) inhibitor, ibrutinib, in B-cell lymphomas, together with the latest approaches that are being developed to overcome this phenomenon.
This is a well written and well-illustrated manuscript that provides a comprehensive and updated view of the research area. The work covers the topic in an objective and analytical manner and includes the most recent developments in the field with a reference list covering the most relevant literature.
Authors have adequately exposed the main point of interest in the field, which consist in 1) the multiple genetic and non-genetic causes for BTKi refractoriness in lymphoid malignancies and 2) the design of new approaches to counteract the different mechanism associated to ibrutinib failure, including rationally-based combination strategies and the newest biological agents.
This reviewer would like to suggest only a few changes, as depicted below:
- Section 2.1: a mention should be made to the relationship between BIRC3 mutation and non-canonical NF-kB signaling as a mechanism of ibrutinib resistance in CLL (Haematologica 2020;105: 448-456) and MCL (Haematologica 2017; 102: e447–e451).
- Section 2.1, line 116: authors associate MYD88L265P mutation with ibrutininb resistance in ABC-DLBCL; however, according to the study by Wilson et al (ref#26), this alteration has been linked to an increased sensitivity to ibrutinib in ABC-DLBCL tumors (80% response). Please correct.
-Section 2.1, lines 128-147: a mention should be made to the first work describing BTKC481S mutation as a mechanism of resistance to ibrutinib in MCL (Cancer Discov 2014;4:1022-35).
- Section 2.2 : authors should include a note about the activation of non-canonical NFκB-signaling after CD40 ligation and the consequent decrease in ibrutinib efficacy in MCL (Cell Death Dis 2018;9:86).
- Section 2.2, lines 160-161: please change “XOP1” for “XPO1”.
- Section 2.2, line 203: please change “check-point” for “checkpoint”.
- Section 4.4: the BRD4i JQ1 and ibrutinib have also shown some synergistic activity, mediated by NF-kB pathway blockade, in ABC-DLBCL (Proc Natl Acad Sci U S A 2014;111:11365-70).
- Table 2 and Table 4: please add a column summarizing the main adverse events associated with each therapeutic regimen.
Author Response
Point 1: Response: As suggested by the reviewer, we have included relationship between BIRC3 mutation and non-canonical NF-kB signaling in line 141-143 and cited respective articles “Importantly, mutation in BIRC3 is associated with activation of non-canonical NF-κB signaling in MCL and CLL which is quite different from the classical NF-κB activation mediated through BCR signaling and therefore this activation of non-canonical NF-κB signaling instructed resistance to ibrutinib” [38,39].
- Vidal-Crespo, A.; Rodriguez, V.; Matas-Cespedes, A.; Lee, E.; Rivas-Delgado, A.; Gine, E.; Navarro, A.; Bea, S.; Campo, E.; Lopez-Guillermo, A., et al. The Bruton tyrosine kinase inhibitor CC-292 shows activity in mantle cell lymphoma and synergizes with lenalidomide and NIK inhibitors depending on nuclear factor-kappaB mutational status. Haematologica 2017, 102, e447-e451, doi:10.3324/haematol.2017.168930.
- Diop, F.; Moia, R.; Favini, C.; Spaccarotella, E.; De Paoli, L.; Bruscaggin, A.; Spina, V.; Terzi-di-Bergamo, L.; Arruga, F.; Tarantelli, C., et al. Biological and clinical implications of BIRC3 mutations in chronic lymphocytic leukemia. Haematologica 2020, 105, 448-456, doi:10.3324/haematol.2019.219550.
Point 2: Response: We agree with reviewer that concept of MYD88L256P associated ibrutinib resistance in line 116 was not very clear and therefore, it may mislead to readers. According to study by Wilson et al (ref#34 in revised manuscript) “Tumours with MYD88 mutations, but wildtype CD79B, were unresponsive to ibrutinib (0/7). The ibrutinib insensitivity of MYD88-only mutant tumours raises the possibility of a MYD88-dependent, but BCR-independent, genetic pathway to ABC-DLBCL”.
We have now corrected this line 116-120 in revised manuscript “Mutations in the MYD88 gene (MYD88L265P) are among the most prevalent in B cell lymphomas including in activating B-cell-like DLBCL (ABC-DLBCL). MYD88L265P mutated ABC-DLBCL tumors with concomitant mutation in BCR signaling component CD79A/B responded to ibrutinib (80% response rate), but tumours harbouring MYD88L265P mutation with wild-type CD79A/B were resistant to ibrutinib, suggesting that these tumours could probably addicted to MYD88-dependent survival signaling [34].
Point 3: Response: As suggested by the reviewer, we have now explained in the article in line 132-134 “By WES, Chiron et al first demonstrated BTKC481S mutation as a mechanism of ibrutinib resistance in relapse MCL tumours, which was however, absent in patients with primary ibrutinib resistance or in those who showed transient response to ibrutinib [36]”.
- Chiron, D.; Di Liberto, M.; Martin, P.; Huang, X.; Sharman, J.; Blecua, P.; Mathew, S.; Vijay, P.; Eng, K.; Ali, S., et al. Cell-cycle reprogramming for PI3K inhibition overrides a relapse-specific C481S BTK mutation revealed by longitudinal functional genomics in mantle cell lymphoma. Cancer Discov 2014, 4, 1022-1035, doi:10.1158/2159-8290.CD-14-0098.
Point 4: Response: Now in this revised manuscript, lines 168-170: “Another report in MCL cell line has shown that the activation of non-canonical NF-κB and MAPK pathway through CD40L-CD40 signaling could bypass BTK signaling consequently led to decrease in ibrutinib efficacy [43-45].
- Rauert-Wunderlich, H.; Rudelius, M.; Berberich, I.; Rosenwald, A. CD40L mediated alternative NFkappaB-signaling induces resistance to BCR-inhibitors in patients with mantle cell lymphoma. Cell Death Dis 2018, 9, 86, doi:10.1038/s41419-017-0157-6.
- Hershkovitz-Rokah, O.; Pulver, D.; Lenz, G.; Shpilberg, O. Ibrutinib resistance in mantle cell lymphoma: clinical, molecular and treatment aspects. Br J Haematol 2018, 181, 306-319, doi:10.1111/bjh.15108.
- Luo, Z.S.a.L. CD40L-CD40 signaling on B-cell lymphoma response to BTK inhibitors. In Proceedings of AACR 107th Annual Meeting
Point 5: Response: We have changed XOP1 to XPO1, lines 170 and 171 in this revised manuscript.
Point 6: Response: We have made changes as suggested by reviewer, now in line, 215 and 217 in revised manuscript.
Point 7: Response: We have updated this information in text (line number 545) and added respective reference in revised manuscript.
Point 8: Response: We have added a new column in table 2 and 4 summarizing the main adverse event associated with each therapeutic regimen.
Reviewer 2 Report
The manuscript by George et al. focuses on strategies to target BTK in various B-cell lymphomas and chronic lymphocytic leukemia (CLL). Ibrutinib, an oral covalent inhibitor of BTK, has shown promising responses in a large number of pre-clinical and clinical studies. However, a frequent primary and acquired resistance to ibrutinib, prompting an urgent need to identify alternative therapeutic options.
This manuscript provides a comprehensive review of (1) basic information about mechanisms of ibrutinib resistance, (2) clinical trials with ibrutinib monotherapy, and combination therapy with different agents, and (3) strategies to overcome ibrutinib resistance, including the third generation of BTK inhibitors.
It is a very informative, well organized, and very well written review. The authors highlighted all the essential references. In my opinion, the manuscript would be of high interest to the readership of CANCERS. However, before publication, a few statements need to be clarified:
- Lines 159-160: “Another report showed that activation of the NF-kB pathway is a critically important step in the development of primary ibrutinib resistance in MCL.” It is misleading because BTK itself activates NK-kB, and ibrutinib-sensitive cells are NF-kB positive. The authors probably meant “the activation of NF-kB downstream of BTK.”
- Line 171 “CLL cells in the lymph node activate BCR signaling.” It should be: “CLL cells in the lymph node have activated BCR signaling.”
- It would be helpful to mention that various cell types in the tumor environment express BTK, and these cells might be affected/suppressed by ibrutinib treatment.
- The authors do not discuss toxicity and various side effects caused by ibrutinib treatment. They only mention about diarrhea and rashes in CLL patients (line 300).
Author Response
Point 1: Response: We agree with reviewer that the above describe text in line “159-160” (now in line 168-170 in revised manuscript) can mislead to readers, therefore, we have modified this sentence in a more specific format and also we have added respective citations.
Lines 168-170: “Another report in MCL cell line has shown that the activation of non-canonical NF-κB and MAPK pathway through CD40L-CD40 signaling could bypass BTK signaling consequently led to decrease in ibrutinib efficacy [43-45].
- Rauert-Wunderlich, H.; Rudelius, M.; Berberich, I.; Rosenwald, A. CD40L mediated alternative NFkappaB-signaling induces resistance to BCR-inhibitors in patients with mantle cell lymphoma. Cell Death Dis 2018, 9, 86, doi:10.1038/s41419-017-0157-6.
- Hershkovitz-Rokah, O.; Pulver, D.; Lenz, G.; Shpilberg, O. Ibrutinib resistance in mantle cell lymphoma: clinical, molecular and treatment aspects. Br J Haematol 2018, 181, 306-319, doi:10.1111/bjh.15108.
- Luo, Z.S.a.L. CD40L-CD40 signaling on B-cell lymphoma response to BTK inhibitors. In Proceedings of AACR 107th Annual Meeting
Point 2: Response: We have changed the sentence as suggested by reviewer. Now line number 181 in revised manuscript.
Point 3: Response: As suggested by reviewer, we have made required changes described in line number 186-187 in this revised manuscript.
Point 4: Response: We have now discussed toxicity and various side effects caused by ibrutinib treatment (line 307-335) in revised manuscript.
“Ibrutinib, although selective towards its target BTK, has many off-targets including interleukin-2-inducible T-cell kinase (ITK), epidermal growth factor receptor (EGFR), and Phosphoinositide 3-kinase (PI3K) [84]. Adverse clinical effects of ibrutinib treatment including diarrhea (60%) and rashes (25%) in CLL patients that often observed are associated to EGFR inhibition and these side-effects are also frequently observed in patients undergoing EGFR targeted therapies [85]. A pooled analysis of four large, randomized, controlled studies included 1505 CLL and MCL patients has shown an increased in incidence of atrial fibrillation in patients treated with ibrutinib therapy with frequency ranging from 4 to 10% above to that is expected in general population [86]. Moreover, ibrutinib therapy is associated with marked increase in incidence of ventricular arrhythmias and sudden cardiac death [87,88]. The underlying mechanism of arrhythmias is not well documented but it could be associated with alleviation in cardiac PI3K/Akt signaling [89]. A multicentric cohort study noted hypertension as a common adverse event associated to patients who were treated with ibrutinib in clinical trial setting with median time to reach peak blood pressure was 6 months (incidence rate of 18%; and for grade ≥3 hypertension in 6%) [90]. Increased risk of minor bleeding is an another ibrutinib treatment associated toxicity observed in up to 66% of patients and this led to serious bleeding complications in patients those were on anticoagulants such as warfarin or aspirin along with ibrutinib treatment [91]. This increased risk of bleeding in patients after ibrutinib treatment is attributed to on-target BTK inhibition as patients with X-linked agammaglobulinemia (absence of BTK expression and defect in platelets aggregation) do not develop excessively bleeding deformities [92]. Arthralgia and myalgia with grade 1-2 has been reported during initial course of ibrutinib treatment in nearly 20% of CLL patients respectively and more severe cases are being rare. So far there is no published data confirm the causative mechanism of arthralgia due to ibrutinib treatment, but this could be because of off-target kinase activity of ibrutinib as this complications are less frequent with other BTK specific inhibitor, acalabrutinib [93]. A pooled meta-analysis from 7 randomized controlled studies of ibrutinib versus comparator included 2167 of patients with CLL, MCL, and WM presented increased risk of upper respiratory tract infection in ibrutinib treated group then in those treated with comparator [94]. There are several mechanism of increased risk of infection in ibrutinib treated patients have been reported. One such mechanism is impairment in immune cell functions such as reduction in natural killer cell antibody-dependent cellular cytotoxicity ability, and phagocytic ability of macrophages [95,96]. Due to the off-target toxicities associated with ibrutinib treatment, nearly 30% of patients who enroll in ibrutinib therapy ultimately discontinue treatment.”
- Barf, T.; Covey, T.; Izumi, R.; van de Kar, B.; Gulrajani, M.; van Lith, B.; van Hoek, M.; de Zwart, E.; Mittag, D.; Demont, D., et al. Acalabrutinib (ACP-196): A Covalent Bruton Tyrosine Kinase Inhibitor with a Differentiated Selectivity and In Vivo Potency Profile. J Pharmacol Exp Ther 2017, 363, 240-252, doi:10.1124/jpet.117.242909.
- Chi, J.; Park, J.; Saif, M.W. Ibrutinib-Induced Vasculitis in a Patient with Metastatic Colon Cancer Treated in Combination with Cetuximab. Case Rep Oncol Med 2020, 2020, 6154213, doi:10.1155/2020/6154213.
- Brown, J.R.; Moslehi, J.; O'Brien, S.; Ghia, P.; Hillmen, P.; Cymbalista, F.; Shanafelt, T.D.; Fraser, G.; Rule, S.; Kipps, T.J., et al. Characterization of atrial fibrillation adverse events reported in ibrutinib randomized controlled registration trials. Haematologica 2017, 102, 1796-1805, doi:10.3324/haematol.2017.171041.
- Guha, A.; Derbala, M.H.; Zhao, Q.; Wiczer, T.E.; Woyach, J.A.; Byrd, J.C.; Awan, F.T.; Addison, D. Ventricular Arrhythmias Following Ibrutinib Initiation for Lymphoid Malignancies. J Am Coll Cardiol 2018, 72, 697-698, doi:10.1016/j.jacc.2018.06.002.
- Salem, J.E.; Manouchehri, A.; Bretagne, M.; Lebrun-Vignes, B.; Groarke, J.D.; Johnson, D.B.; Yang, T.; Reddy, N.M.; Funck-Brentano, C.; Brown, J.R., et al. Cardiovascular Toxicities Associated With Ibrutinib. J Am Coll Cardiol 2019, 74, 1667-1678, doi:10.1016/j.jacc.2019.07.056.
- McMullen, J.R.; Boey, E.J.; Ooi, J.Y.; Seymour, J.F.; Keating, M.J.; Tam, C.S. Ibrutinib increases the risk of atrial fibrillation, potentially through inhibition of cardiac PI3K-Akt signaling. Blood 2014, 124, 3829-3830, doi:10.1182/blood-2014-10-604272.
- Roeker, L.E.; Sarraf Yazdy, M.; Rhodes, J.; Goodfriend, J.; Narkhede, M.; Carver, J.; Mato, A. Hypertension in Patients Treated With Ibrutinib for Chronic Lymphocytic Leukemia. JAMA Netw Open 2019, 2, e1916326, doi:10.1001/jamanetworkopen.2019.16326.
- Shatzel, J.J.; Olson, S.R.; Tao, D.L.; McCarty, O.J.T.; Danilov, A.V.; DeLoughery, T.G. Ibrutinib-associated bleeding: pathogenesis, management and risk reduction strategies. J Thromb Haemost 2017, 15, 835-847, doi:10.1111/jth.13651.
- Nicolson, P.L.R.; Hughes, C.E.; Watson, S.; Nock, S.H.; Hardy, A.T.; Watson, C.N.; Montague, S.J.; Clifford, H.; Huissoon, A.P.; Malcor, J.D., et al. Inhibition of Btk by Btk-specific concentrations of ibrutinib and acalabrutinib delays but does not block platelet aggregation mediated by glycoprotein VI. Haematologica 2018, 103, 2097-2108, doi:10.3324/haematol.2018.193391.
- Stephens, D.M.; Byrd, J.C. How I manage ibrutinib intolerance and complications in patients with chronic lymphocytic leukemia. Blood 2019, 133, 1298-1307, doi:10.1182/blood-2018-11-846808.
- Ball, S.; Das, A.; Vutthikraivit, W.; Edwards, P.J.; Hardwicke, F.; Short, N.J.; Borthakur, G.; Maiti, A. Risk of Infection Associated With Ibrutinib in Patients With B-Cell Malignancies: A Systematic Review and Meta-analysis of Randomized Controlled Trials. Clin Lymphoma Myeloma Leuk 2020, 20, 87-97 e85, doi:10.1016/j.clml.2019.10.004.
- Borge, M.; Belen Almejun, M.; Podaza, E.; Colado, A.; Fernandez Grecco, H.; Cabrejo, M.; Bezares, R.F.; Giordano, M.; Gamberale, R. Ibrutinib impairs the phagocytosis of rituximab-coated leukemic cells from chronic lymphocytic leukemia patients by human macrophages. Haematologica 2015, 100, e140-142, doi:10.3324/haematol.2014.119669.
- Holbrook E Kohrt, S.E.M.H., Jonathan P Butchar, Carolyn Cheney, Xiaoli Zhang, Joseph J. Buggy, Natarajan Muthusamy, Ronald Levy, Amy J Johnson, John C. Byrd. Ibrutinib (PCI-32765) Antagonizes Rituximab-Dependent NK-Cell Mediated Cytotoxicity. In Proceedings of ASH Publicayion Blood.
Reviewer 3 Report
The manuscript is well written and comprehensive on Ibrutinib resistance mechanisms. It is acceptable for publication and will be helpful for clinicians.
Author Response
Response: We are pleased to know that reviewer found this review article will have value and will be helpful for clinicians. To further improve this manuscript, we have made additional changes as suggested by reviewer 1 and reviewer 2.